# HATFormer: Historic Handwritten Arabic Text Line Recognition with Transformers

## Abstract

Arabic handwritten text recognition (HTR) is challenging, especially for historical texts, due to diverse writing styles and the intrinsic features of Arabic script. Additionally, Arabic handwriting datasets are smaller compared to English ones, making it difficult to train generalizable Arabic HTR models. To address these challenges, we propose HATFormer, a transformer-based encoder–decoder architecture that builds on a state-of-the-art English HTR model. By leveraging the transformer's attention mechanism, HATFormer captures spatial contextual information to address the intrinsic challenges of Arabic script through differentiating cursive characters, decomposing visual representations, and identifying diacritics. Our customization to historical handwritten Arabic includes an image processor for effective ViT information preprocessing, a text tokenizer for compact Arabic text representation, and a training pipeline that accounts for a limited amount of historic Arabic handwriting data. HATFormer achieves a character error rate (CER) of 8.6% on the largest public historical handwritten Arabic dataset, with a 51% improvement over the best baseline in the literature. HATFormer also attains a comparable CER of 4.2% on the largest private non-historical dataset. Our work demonstrates the feasibility of adapting an English HTR method to a low-resource language with complex, language-specific challenges, contributing to advancements in document digitization, information retrieval, and cultural preservation.

## 1 Introduction

Global archives contain hundreds of millions of manuscript pages written in the Arabic alphabet, primarily from the 19th and early 20th centuries, with around 25 million images from the Middle East and North Africa alone. For historians, the laborious process of sifting through these pages for relevant data is impractical due to time and resource constraints. Existing handwritten text recognition (HTR) systems for non-historical Arabic texts fail to effectively render these historical documents into a searchable format. Developing a dedicated HTR system for historical Arabic manuscripts would revolutionize digital humanities, enabling rapid data search and retrieval while facilitating the creation of advanced large language models for research, thus opening new avenues for historical and humanities scholarship.

This paper introduces HATFormer, a transformer-based historical Arabic HTR system that leverages self-attention mechanisms to capture long-range dependencies, outperforming traditional HTR methods for complex scripts like Arabic. HTR systems such as Shi et al. (2016) have traditionally relied on convolutional neural networks (CNNs) (LeCun et al., 1989) for feature extraction and recurrent neural networks (RNNs) (Rumelhart et al., 1986) for text generation. However, RNN-based methods often struggle to capture long-range dependencies, which are more crucial for handling Arabic scripts than for English scripts. Recently, transformer (Vaswani et al., 2017) methods have shown to be promising for modern and historical English HTR tasks, with Li et al. (2023), Fujitake (2024), and Parres & Paredes (2023) achieving state-of-the-art character error rates (CER) of 2.9%, 2.4%, and 2.7%, respectively. HATFormer builds on the success of pretrained vision and text transformers in HTR, introducing key adaptations to handle the intrinsic challenges of Arabic for more accurate recognition of historical text.

We will show through experimental verification that the inductive bias of the transformer's attention mechanism effectively addresses the following three intrinsic challenges (Najam & Faizullah, 2023; Faizullah et al., 2023) of Arabic script absent in English. First, Arabic is required to be written in cursive, making characters visually harder to distinguish. The attention mechanism allows the model to better differentiate between connected characters. Second, Arabic characters are context-sensitive, meaning a character's shape can change depending on its position in a word and adjacent characters. Attention helps accurately decompose these visual representations by focusing on the relevant context within the sequence. Third, the Arabic language includes diacritics, which are markings above or below characters that can completely alter the semantics of a word. Attention enables the model to effectively identify diacritics by considering their contextual influence on surrounding characters.

In addition to the intrinsic challenges of Arabic scripts, Arabic handwritten datasets, especially historical ones, are significantly smaller than those available for languages like English. Many HTR works (Li et al., 2023; Wigington et al., 2018; Zhang et al., 2019) focus on languages using the modern Latin alphabet, such as English and French, where large amounts of training data are readily available. Common datasets include IAM (Marti & Bunke, 2002), with over 1,500 handwritten pages, and RIMES (Grosicki et al., 2024), which comprises a mix of handwritten and printed text across approximately 12,500 pages. In contrast, the largest public dataset for handwritten Arabic (Saeed et al., 2024) consists of just over 1,600 pages, while another widely used dataset, KHATT (Mahmoud et al., 2012), contains only 1,000 pages. A notable exception is the MADCAT (Lee et al., 2012; 2013a;b) dataset, which contains over 40,000 pages of handwritten Arabic. However, it is not focused on historical writing, highlighting the limited availability of resources for historical texts.

We base our approach on TrOCR (Li et al., 2023) and leverage domain knowledge of the Arabic language to identify key factors in building an effective historical Arabic HTR system. We incorporate a novel image preprocessor and synthetic dataset generator to enhance performance by minimizing horizontal information loss and expanding the training dataset with realistic synthetic images. We perform extensive evaluation and cross-dataset experiments on HATFORMER. We will release the image preprocessor, tokenizer, model weights, and source code for our HTR system, along with a detailed guide for researchers to interface our system with existing text detection packages for page-level HTR evaluations and practical deployment. Additionally, we will release our dataset of realistic synthetic Arabic images and its generation source code, as well as provide an OCR Error Diagnostic App and its source code to benefit both machine learning and history studies researchers. The contributions of our work are threefold.

1. Our proposed HATFORMER for historical Arabic HTR outperforms the state of the art across various Arabic handwritten datasets. It achieves a CER of 8.6% and 4.2% on the largest public and private handwritten Arabic datasets, respectively.

2. Our method has proven effective by leveraging the attention mechanism to address three intrinsic challenges of the Arabic language.

3. Our historical Arabic HTR system and OCR Error Diagnostic App will aid humanity researchers by automatically transcribing historical Arabic documents and debugging common recognition errors, thereby significantly enhancing the accessibility of these documents.

## 2 RELATED WORKS

**Handwritten Text Recognition (HTR).** Handcrafted features were historically used for optical character recognition (OCR) and HTR (Balm, 1970), but deep learning methods gradually took over due to their improved performance. Common deep learning methods adopt the encoder–decoder paradigm where visual signals are encoded into a feature representation and the feature is decoded for text generation. Graves & Schmidhuber (2008) proposed using a long short-term memory (LSTM) (Hochreiter & Schmidhuber, 1997) multidimensional recurrent neural network (MDRNN) (Graves et al., 2007) for feature extraction and a connectionist temporal classification (CTC) layer for decoding (Graves et al., 2006). Notably, Shi et al. (2016) introduced the convolutional recurrent neural network (CRNN) architecture for OCR, where a CNN was used to extract visual features from images, and a stacked bidirectional LSTM (BLSTM) (Graves & Schmidhuber, 2005; Graves et al., 2013) was used as the decoder. Puigcerver (2017); Wang & Hu (2017) respectively adapted the encoder to use a CNN and modified recurrent convolutional neural network (RCNN) (Liang & Hu, 2015). Newer approaches (Michael et al., 2019; Wang et al., 2020)

attempt to incorporate the attention mechanism (Bahdanau et al., 2015) into the HTR pipeline. Coquenet et al. (2023) use the attention mechanism to perform full-page HTR, bypassing the need for line-level segmentation.

**Transformers for HTR.** Transformers (Vaswani et al., 2017) have recently been applied to HTR with earlier works using architectures consisting of a CNN-feature-extractor encoder and a transformer-encoder–decoder-hybrid decoder, which later was simplified to a transformer-only encoder–decoder pair or a transformer-decoder-only architecture. Wick et al. (2021) proposed a hybrid system that uses a CNN feature extractor and multiple encoder–decoder transformers for bidirectional decoding. Li et al. (2023) proposed a transformer-only method utilizing pretrained vision transformers (ViT) (Dosovitskiy et al., 2021), specifically BEiT (Bao et al., 2022), as its encoder using raw pixels as input and text transformers, specifically RoBERTa (Liu et al., 2019), as the decoder. Fujitake (2024) proposed a transformer-decoder-only method, using GPT (Radford et al., 2018; 2019) in particular, with raw pixel inputs. However, the decoder-only method, in general, requires more labeled data for training end-to-end, whereas a pretrained encoder could be used as an initialization step for visual feature extraction. Our approach does not use a dedicated CNN feature extractor and builds upon the transformer-only encoder–decoder architecture. ViTs have been shown to outperform CNNs and can benefit from large-scale pretraining for downstream tasks with low resources (Dosovitskiy et al., 2021), like Arabic HTR.

**Arabic HTR.** Arabic handwriting poses unique challenges to HTR systems, such as cursive writing, connected letters, and context-dependent character shapes. One of the earliest approaches to Arabic HTR is Graves & Schmidhuber (2008), which proposes using multidimensional LSTM (MDLSTM) and CTC decoding. Shtaiwi et al. (2022); Lamtougui et al. (2023); Saeed et al. (2024) proposed using a CNN and BLSTM architecture, with Shtaiwi et al. (2022); Saeed et al. (2024) based upon the Start, Follow, Read network (Wigington et al., 2018). As with traditional English HTR, many Arabic HTR systems are starting to use the transformer architecture. Mostafa et al. (2021) proposed a method that combines a ResNet-101 (He et al., 2016) for feature extraction and an encoder–decoder transformer for text prediction. Momeni & BabaAli (2024) proposed a system that solely uses transformers, similar to Li et al. (2023), but also introduces transducers (Graves, 2012) for HTR, removing the need for external postprocessing language models. We continue using transformers for HTR and leverage the most recent advancements to further improve recognition performance on Arabic texts.

## 3 BACKGROUND AND PRELIMINARIES

This section provides background information about the components that HATFORMER is built on.

**Arabic-Character Encoding.** Arabic characters can be efficiently represented in tokens for learning using byte-level byte pair encoding (BBPE) (Radford et al., 2019). It is a tokenization technique that compresses a string into a reversible compact representation by leveraging the UTF-8 encoding standard using a vocabulary dictionary. To train the vocabulary dictionary, it is initialized with all 256 possible byte values as base tokens, allowing it to tokenize any Unicode character and eliminating the need for a task-specific vocabulary. They are then iteratively merged based on the most frequent token pairs in a corpus to form new tokens. This iterative process expands the vocabulary, allowing for more efficient encoding of frequent patterns.

**TrOCR.** The TrOCR framework (Li et al., 2023) for predicting text from images will be used as the base architecture for this work. TrOCR employs a transformer-only encoder–decoder architecture, specifically using a pretrained ViT as the encoder and a pretrained text transformer as the decoder. The encoder takes an input image of shape $3 \times H_0 \times W_0$, which is resized to a fixed shape of $3 \times H \times W$. The resized image is then decomposed into a sequence of $N = HW/P^2$ patches, where each patch has a shape of $3 \times P \times P$. The encoder will use the patches with added positional embeddings as input to generate encoder embeddings. The decoder employs masked attention on the ground-truth text tokens to ensure it does not access more information during training than during prediction. The ground-truth text tokens are then combined with the encoder embedding using cross-attention. A linear layer projects the hidden states from the decoder to match the vocabulary size, and the probabilities over the vocabulary are computed using the softmax function. Beam search is used to generate the final output.

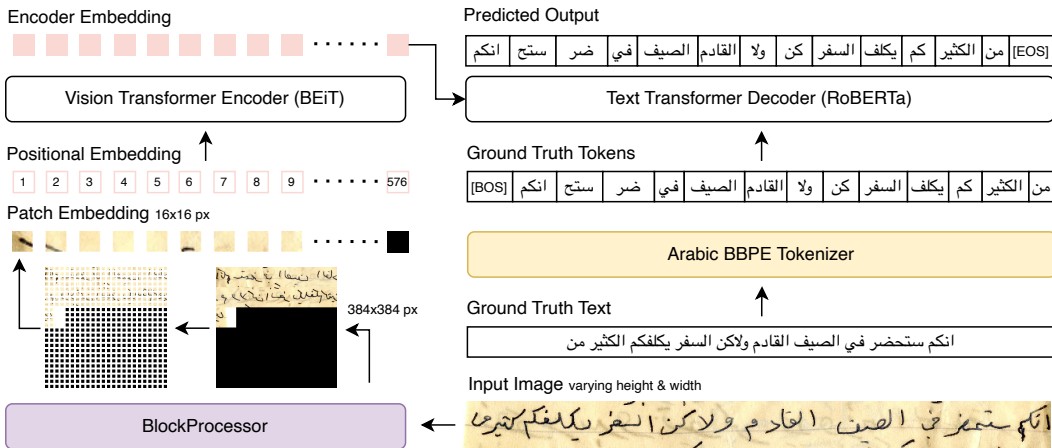

Figure 1: The architecture of HATFORMER. The input text-line image is processed by our BLOCK-PROCESSOR and the BEiT vision transformer. The ground-truth text string is tokenized using our Arabic BBPE tokenizer. The RoBERTa transformer is used for text prediction. HATFORMER addresses the three intrinsic challenges of Arabic scripts by leveraging attention and is able to work on smaller datasets with the help of our synthetic image training pipeline.

## 4    PROPOSED METHOD FOR HISTORICAL ARABIC HTR

In this section, we present HATFORMER, which tackles the unique challenges of Arabic handwriting recognition, particularly for historical documents. We describe the main components of our method, including an image processor for effective ViT information preprocessing, a text tokenizer for compact Arabic text representation, and a training pipeline that accounts for the limited availability of historic Arabic handwriting data.

**Architecture and Unit of Analysis.** Prediction for HTR involves recognizing and converting a text image into machine-readable characters. As illustrated in Figure 1, HATFORMER follows TrOCR's transformer encoder–decoder architecture for HTR text prediction. We focused on line-level images as in Li et al. (2023); Momeni & BabaAli (2024), which is more challenging than the word- and character-level predictions but less complex than the paragraph- and page-level predictions. This approach allows us to focus on Arabic HTR without the additional complexities of text document structure. HATFORMER can be easily integrated with existing layout detection methods, enabling full-page prediction capabilities.

### 4.1    BLOCKPROCESSOR FOR EFFECTIVE VIT INFORMATION PREPROCESSING

We introduce a BLOCKPROCESSOR to best prepare each text-line image for effective ViT comprehension by applying image-processing insights and leveraging ViT's blocking and indexing behaviors. The proposed BLOCKPROCESSOR works by first horizontally flipping a text-line image, then standardizing its height to 64 pixels, and finally warping it to fill in the ViT's 384×384-pixel image container from left to right and top to bottom. The ViT's image container will allow up to six nonoverlapping complete rows that are 384 pixels wide, accommodating line images with varying widths for up to 2,304 pixels. For shorter images, zeros will be padded. Figure 2(c) shows an output of the proposed BLOCKPROCESSOR respecting the input image's aspect ratio to allow potential perfect reconstruction. In contrast, images in Figure 2(d), (b), and (f) show significant information loss due to the direct use of ViT's image preprocessor. We provide analysis below and justify the system design of BLOCKPROCESSOR.

**Aspect Ratio.** ViT resizes input images to 384×384 without respecting their original aspect ratios. This leads to an inefficient representation of text-line images from the Muharaf dataset, which has an average image width of 614 pixels after standardizing their heights to 64 pixels. A direct application of ViT image preprocessing will lead to horizontal compression of 1.6 times on average, losing the clarity of the strokes in the horizontal direction for confident recognition. Figure 2(d) shows a

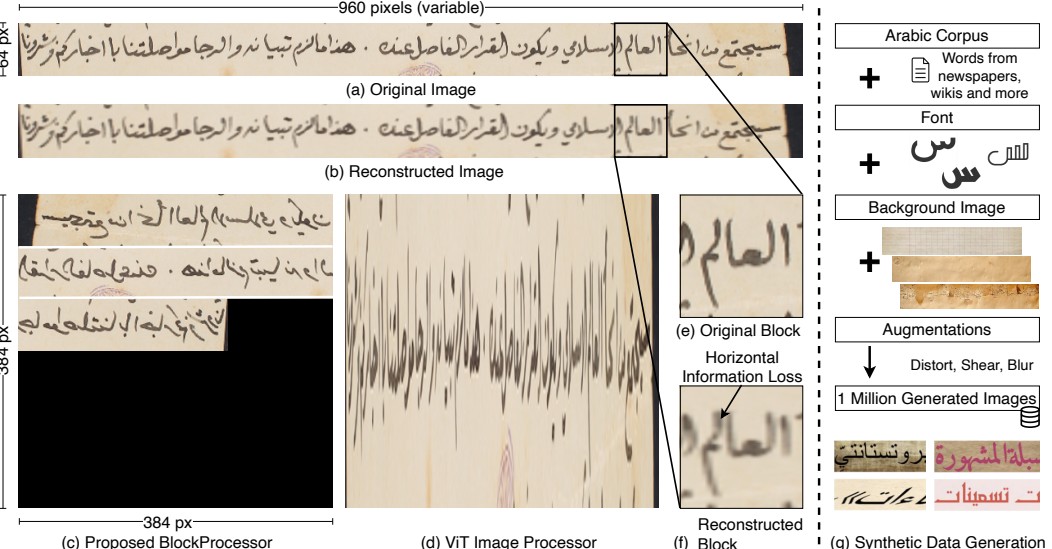

Figure 2: **Left**: Our proposed BLOCKPROCESSOR respects the aspect ratio of (a) an original image and chunks it to fit within (c) a 384×384-pixel ViT image container. In contrast, the base ViT image processor naively resizes images to (d) fully occupy its square image container, resulting in (f) significant horizontal information loss of the vertical strokes when compared to (e) the raw version. **Right**: (g) Synthetic image generation pipeline. Realistic-looking text-line images are generated by randomly selecting words from a large Arabic corpus, rendering with a random font, paper background, and image augmentation.

ViT-resized text-line image and Figure 2(b) shows the image content if the resized image is rescaled back to its original shape. As the zoomed-in reconstruction block in Figure 2(f) reveals, the vertical strokes suffer the most severe blurring, making it difficult for any observer to confidently determine the exact thickness of a stroke at different vertical heights.

**ViT Blocking & Indexing.** In the BLOCKPROCESSOR, both horizontal flipping of text-line images and the standardization of their heights to 64 pixels are designed to better leverage ViT's blocking and indexing behaviors for more efficient transformer training. First, Arabic text-line images read from right to left, so flipping them horizontally can avoid representing the end of a sentence with beginning ViT image tokens. Even though positional embeddings will help with ordering, we opt not to add extra workload to the attention layers as it will potentially require more training data. Second, we standardize the line image to an integer multiple of the ViT's patch height of 16 pixels. This resizing choice ensures that when Arabic texts are written in the middle of a text line, the corresponding ViT image tokens with foreground text will always have similar indices. Without resizing the height to an integer multiple of 16 pixels, the boundary of a text line and the boundary of a ViT block will misalign at varying pixel counts for different rows. This will cause foreground text to appear in all ViT image tokens, increasing the learning complexity for the attention layers.

## 4.2 ARABIC BBPE TEXT TOKENIZER FOR COMPACT ARABIC TEXT REPRESENTATION

Text representation is integral for language modeling. Radford et al. (2019) showed the impact of using a byte-level representation for text with byte-level byte pair encoding (BBPE). This led to an optimal balance of token sequence length and vocabulary size. To efficiently represent Arabic text, we trained our own custom BBPE dictionary on a combined corpus from Abbas & Smaili (2005); Abbas et al. (2011); Saad & Alijla (2017). As the base BBPE dictionary from Radford et al. (2019) is skewed toward ASCII characters, our experiments show that Arabic text is represented with over 300% more tokens compared to our custom BBPE dictionary. The more compact representation from the custom BBPE dictionary results in a less complicated classification problem, resulting in higher accuracy along with our BLOCKPROCESSOR, as we will discuss in Section 5.5.

### 4.3 TRAINING ON REALISTIC SYNTHETIC AND REAL-WORLD TEXTLINE IMAGES

Our proposed method involves a two-stage training/fine-tuning process, i.e., training on a large synthetic dataset followed by fine-tuning on a real-world Arabic handwritten dataset.

**Stage 1–Training on Large Synthetic Printed Dataset.** To address the scarcity of historical handwritten Arabic data and capture key intrinsic features of Arabic scripts, we first trained HAT-FORMER on a large dataset of one million synthetic text-line images. This approach mitigates the impact posed by the limited availability of historical handwritten Arabic data. The synthetic images contain all three inherent characteristics of Arabic, i.e., cursive writing, context-dependent character shapes, and diacritics. This enables our system to learn these challenging characteristics of Arabic script before being trained on a downstream task. Synthetic training provides the necessary data for the encoder to learn the visual features of Arabic, leading to more effective generalization. The synthetic image generation pipeline will be described in Section 5.1 and shown in Figure 2.

**Stage 2–Fine-Tuning on Real Handwritten Dataset.** We fine-tune HATFORMER on real Arabic handwritten datasets, primarily focusing on the Muharaf dataset containing 36,000 text-line images due to its relevance to historical handwritten texts. To achieve strong performance on Arabic HTR, we leverage a technical insight for large-scale training from Hao et al. (2019); Mosbach et al. (2021). Traditional machine learning theory suggests that when the validation loss flattens, the model has converged, and no further learning occurs (Mohri et al., 2018; Jo, 2021). However, Mosbach et al. (2021) demonstrated that transformers can continue to improve in task performance long after the validation loss has plateaued. Mosbach et al. (2021) indicates that achieving a near-perfect training loss can serve as a strong baseline for model performance. In Stage 2, we train past the plateau of the validation loss and approach a near-perfect training loss while monitoring the validation CER as the stopping criteria, which can take twice as long as the minimum validation loss.

## 5 EXPERIMENTAL RESULTS

We present the experimental results for HATFORMER on three Arabic handwritten datasets and compare it with other Arabic HTR baselines. We also conduct ablation studies to assess the effectiveness of each component and analyze various parameters of HATFORMER.

### 5.1 SYNTHETIC & REAL-WORLD ARABIC DATASETS

**Synthetic Stage 1 Training Dataset.** For our Stage 1 training dataset, we generated 1,000,065 synthetic images of Arabic text lines. We first randomly sampled between 1–20 words from an Arabic corpus containing 8.2 million words. The sampled words were then paired with one of 54 Arabic text fonts on a background chosen from 130 paper background images and one of eight image augmentations to generate synthetic line images. Our ablation study in Section 5.5 will show that English OCR initialization is insufficient and synthetic Arabic training is required.

**Arabic HTR Datasets.** The Muharaf dataset (Saeed et al., 2024) is a collection of historical handwritten Arabic manuscripts that span from the early 19th century to the early 21st century. The dataset contains over 36,000 text line images, which vary significantly in quality, from clear writing on clean white backgrounds to illegible sentences on creased pages with ink bleeds. The KHATT dataset (Mahmoud et al., 2012) is a collection of Arabic handwriting samples with over 6,600 segmented line images. All images have black text on a clean white background. The MADCAT dataset (Lee et al., 2012; 2013a;b) is a collection of 740,000 handwritten Arabic line images created under controlled writing conditions. All images have black text on a clean, white background. See Appendix B for more detailed descriptions of each dataset.

### 5.2 EXPERIMENTAL CONDITIONS

We initialized our model from HuggingFace's `trocr-base-stage1` 334M parameter model. We use BEiT (Bao et al., 2022) and RoBERTa (Liu et al., 2019) as the encoder and decoder, respectively, since TrOCR (Li et al., 2023) empirically showed that they achieved the best CER performance. We used a batch size of 15 with a learning rate of $5 \times 10^{-5}$ and linear warmup of 20,000 steps for synthetic Stage 1 training. For Stage 2 fine-tuning, we used a batch size of 30 with a learn-

Table 1: Performance on Arabic Handwritten Datasets.

| Dataset | Model | Architecture | CER (%) ↓ |
|---------|-------|--------------|-----------|
| Muharaf (Full) | Saeed et al. (2024) | CRNN | 14.9 |
| | Proposed Model | Transformer | **11.7** |
| Muharaf (Arabic Only) | Saeed et al. (2024) | CRNN | 17.6 |
| | Proposed Model | Transformer | **8.6** |
| KHATT | Saeed et al. (2024) | CRNN | **14.1** |
| | Lamtougui et al. (2023) | CRNN | 19.9 |
| | Momeni & BabaAli (2024) | Transformer | 18.5 |
| | Proposed Model | Transformer | 15.4 |
| MADCAT | Saeed et al. (2024) | CRNN | 5.5 |
| | Shtaiwi et al. (2022) | CRNN | 4.0 |
| | Rawls et al. (2018) | CRNN | **1.5**[1] |
| | Proposed Model | Transformer | 4.2 |
| Combined | Proposed Model | Transformer | **15.3** |

[1] Used the 2013 NIST OpenHART evaluation tools for computing CER/WER, which involved normalizing certain diacritics.

Table 2: Cross-Dataset Evaluation.

| Training Data | Test Data | Model | CER (%) ↓ |
|---------------|-----------|-------|-----------|
| Muharaf (Full) | KHATT | Saeed et al. (2024) | 38.5 |
| | | Proposed Model | **22.8** |
| | MADCAT | Saeed et al. (2024) | 30.5 |
| | | Proposed Model | **21.6** |
| Muharaf (Arabic Only) | KHATT | Saeed et al. (2024) | 33.0 |
| | | Proposed Model | **27.5** |
| | MADCAT | Saeed et al. (2024) | 28.9 |
| | | Proposed Model | **26.5** |
| KHATT | Muharaf | Saeed et al. (2024) | 43.8 |
| | | Proposed Model | **40.7** |
| | MADCAT | Saeed et al. (2024) | **17.8** |
| | | Proposed Model | 18.1 |
| MADCAT | Muharaf | Saeed et al. (2024) | 43.5 |
| | | Proposed Model | **41.4** |
| | KHATT | Saeed et al. (2024) | 17.8 |
| | | Proposed Model | **16.3** |

ing rate of $10^{-4}$ and linear warmup of 2,000 steps. The warmup was followed by an inverse square root schedule for both Stage 1 training and Stage 2 fine-tuning. We trained on 2 to 4 A100 or H100 GPUs.

We did Stage 1 training on a train–validation–test dataset split of 90–9–1 for 1,000,065 synthetic line images. We fine-tuned using a split of 85–15–5 for 25,767 line images from the Muharaf dataset; the author recommended a 72–14–14 split for 6,687 line images from the KHATT dataset and a 72–18–10 split for 741,877 line images from the MADCAT dataset. We used traditional validation loss as the early stopping criterion during Stage 1 of training, with a maximum of 5 epochs. However, we used the overtraining technique during our Stage 2 fine-tuning and utilized the validation CER for early stopping as explained in Section 4.3.

## 5.3 MAIN RESULTS

We compare the performance of HATFORMER against state-of-the-art baselines across the Muharaf, KHATT, and MADCAT datasets. We evaluate the HTR performance using the character error rate (CER) (Levenshtein, 1966), which is widely used for assessing the accuracy of OCR and HTR systems (Neudecker et al., 2021). It is defined as CER $= (S + D + I)/N$, where $S$ is the number of substitutions, $D$ is the number of deletions, $I$ is the number of insertions, and $N$ is the total number of characters in the original text. The CER is based on the edit distance, which calculates the number of aforementioned operations required to transform the predicted text into the original text. We also performed cross-dataset comparisons to evaluate HATFORMER's ability to generalize across different datasets.

Table 1 reports the CER for HATFORMER and several existing baselines across the three datasets. An important note is that the only existing baseline for the Muharaf dataset is Saeed et al. (2024). Since the source code for many existing Arabic HTR baseline models is not publicly available, except Saeed et al. (2024), we compared our results to the reported numbers obtained from their papers. For Saeed et al. (2024), we retrained their model on each dataset and with stage-1 synthetic training for a fair comparison. It is important to note that the dataset splits used in these baselines may differ from those in our experiments, potentially affecting direct comparisons. Additionally, we conducted experiments on two variants of Muharaf, the entire dataset and a subset containing only Arabic characters. This allows us to investigate the impact of non-Arabic characters on HTR performance. For clarity, our analysis will refer to the Arabic-only subset as Muharaf.

We first compared with CNN and RNN-based methods. HATFORMER achieves a CER of 8.6% and 15.4% on the Muharaf and KHATT datasets, respectively, as compared to Saeed et al. (2024) who achieved a CER of 17.6% and 14.1%. Lamtougui et al. (2023) achieved a CER of 19.9% on the KHATT dataset. These results indicate that the transformer architecture can significantly outperform traditional HTR methods based on CRNNs with a 23–51% improvement in CER for handwritten

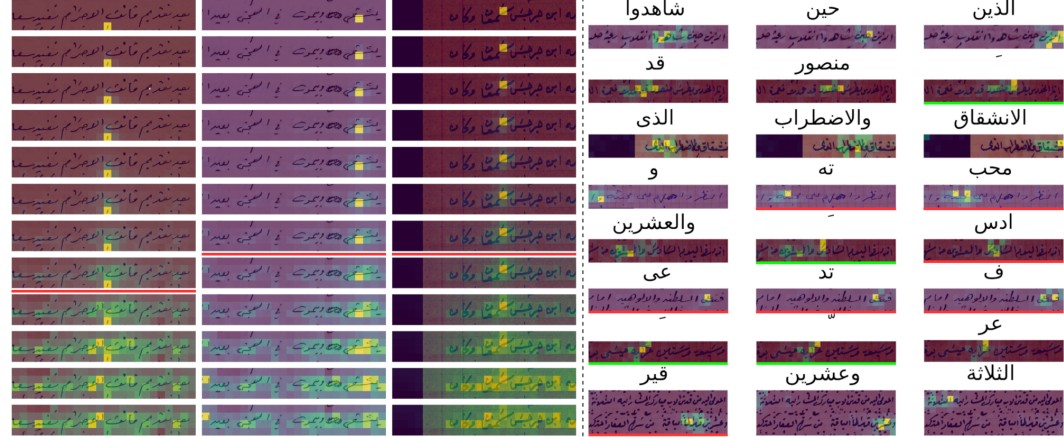

Figure 3: Self- and Cross-attention map visualizations. Yellow highlights areas of greater attention, with attention maps overlaid onto the input image for easier comparison. **Left**: ViT encoder self-attention maps for selected patch tokens. The top of each column shows the relevant patch, followed by attention maps showing what the transformer attends to as it progresses through its subsequent layers. The leftmost column shows the attention for a diacritic patch. Red lines indicate the layer cutoff where the attention association becomes too broad, as identified by our Arabic expert. **Right**: RoBERTa decoder cross-attention maps for selected ground truth text tokens. Each row represents consecutive text tokens, read from right to left, with the decoded token string above each map. Tokens are annotated based on their type: red underlines indicate diacritic tokens, green underlines denote subword tokens, and all other tokens correspond to full words, as identified by our Arabic language expert. The attention maps reveal the model's ability to attend to relevant image regions for each token. It can handle a diverse range of text, from small diacritics to complex compounded characters, demonstrating the model's ability to overcome the inherent challenges of Arabic script.

Arabic. This aligns with computer vision and natural language processing trends, where transformers are increasingly favored due to their superior ability to take care of contextual information.

While Saeed et al. (2024) slightly outperformed our method on the KHATT dataset, HATFORMER remains comparable. We attribute this to differences in the dataset characteristics, which may favor Saeed et al. (2024)'s hybrid architecture. Our results imply that CNNs and RNNs are no longer required for HTR. We enable a fully transformer-based model that can surpass these hybrid architectures by utilizing vision transformers as standalone feature extractors combined with a text transformer decoder. We credit the effectiveness of our model to the attention mechanism, which allows for learning contextual information critical for language modeling. Figure 3 illustrates how the attention mechanism captures character relationships. See Appendix A for a more detailed description and analysis of the attention maps.

We also compared our approach with transformer-based methods. HATFORMER achieves a CER of 15.4%, compared with Momeni & BabaAli (2024), who achieved a CER of 18.5% on the KHATT dataset. Our 17% improvement in CER demonstrates the effectiveness of our preprocessing and overtraining methods. Our preprocessing pipeline mitigates information loss caused by horizontal image compression, resulting in a CER improvement discussed in Section 5.5, while our overtraining strategy establishes a strong baseline, ultimately leading to better performance.

HATFORMER achieves a CER of 4.2%, comparable to other baseline Arabic HTR models on the MADCAT dataset. The 1.5% CER achieved by Rawls et al. (2018) may be due to several factors, specifically text normalization during evaluation (Rawls et al., 2018), which can significantly improve performance as shown in Section 5.5. MADCAT also presents many unique dataset-specific complexities and requires distinct preprocessing techniques, as highlighted by Abandah & Al-Hourani (2018). Furthermore, as both KHATT and MADCAT are non-historical datasets, they pose a different set of challenges compared to the historical Arabic texts that are the main focus of our work. While we included MADCAT and KHATT in our evaluation for a more complete comparison with existing Arabic HTR systems, we did not specifically optimize for them, as our primary goal is to enhance the performance of historical Arabic HTR.

Table 3: Ablation Study on Muharaf.

| | Model | CER (%) ↓ |
|---|---|---|
| (A) | Proposed Model | 8.6 |
| (B) | (A) - Overtraining | 9.9 |
| (C1) | (B) - BLOCKPROCESSOR + TrOCR Processor | 11.4 |
| (C2) | (B) - Modified text tokenizer + TrOCR Tokenizer | 10.0 |
| (D) | (B) - (C1) - (C2) | 10.4 |
| (E) | (D) - Synthetic Stage-1 fine-tuning | 14.6 |
| (F) | (E) - Pretrained weights | 86.0 |

Table 4: Effects of Arabic normalization postprocessing on Muharaf.

| Model | CER (%) ↓ |
|---|---|
| Best Base Model | 8.6 |
| + Remove diacritics | 8.0 |
| + Remove without context | 7.4 |
| + Remove with context | 6.7 |

We combined the three handwritten datasets into a single large dataset to evaluate the model's performance across diverse handwriting styles. Using this combined dataset, HATFORMER achieved a CER of 15.3%. While this result is slightly worse than the individual dataset CERs, it reflects the challenge of adapting to significant image content and style variability across the Muharaf, KHATT, and MADCAT datasets, indicating that HATFORMER can still extract meaningful shared features even with the increased difficulty of combining datasets.

## 5.4 CROSS-DATASET EVALUATION

We conducted cross-dataset evaluations to explore the generalization ability of our model. Table 2 shows the results of cross-dataset evaluation. This table reveals the importance of using historical handwriting data for a strong general Arabic HTR model. While training on modern Arabic handwriting using either KHATT or MADCAT gives a high CER of 40% on Muharaf, training on historical Muharaf data gives a lower CER of 26% on modern Arabic handwriting. Hence, this shows that our model can perform well on the historic Muharaf handwriting and generalize to in-the-wild, unseen modern handwritten Arabic.

We also ran cross-dataset evaluations using Saeed et al. (2024)'s HTR system. As seen in Table 2, our proposed model outperforms their approach in every evaluation except one. Notably, when training on Muharaf (Arabic only) and testing on KHATT, our model outperforms (Saeed et al., 2024) by 16.7% at a CER of 27.5% compared to 33%, which further shows our model's capability to generalize better compared to other methods.

## 5.5 ABLATION STUDY

In our ablation study, we quantify the impact of each component of our model by starting with our best model and removing each component one at a time, as shown in Table 3 .

**Baseline Model (A).** Our baseline model achieved a CER of 8.6%. This served as the benchmark against which we compared the performance of the ablated models.

**Overtraining (B).** When we only trained to the minimum validation loss, we observed a slight increase in CER by 1.3%. This result is consistent with Hao et al. (2019); Mosbach et al. (2021), suggesting that our model was not fully trained.

**BLOCKPROCESSOR and Modified Text Tokenizer. (C1) & (C2) & (D).** When the BLOCKPROCESSOR and Arabic BBPE were added together, this led to a 0.5% CER improvement supporting our ideas in Sections 4.1 and 4.2. Replacing TrOCR's image processor with our BLOCKPROCESSOR led to a 0.4% CER improvement, whereas replacing the modified text tokenizer with TrOCR's tokenizer led to a −1.0% CER performance change. This indicates that the BLOCKPROCESSOR enhances image feature extraction. However, the modified text tokenizer struggles when paired with TrOCR's processor due to the naive resizing, which discards essential features needed for predicting compact Arabic token representations, as discussed in Section 4.1. The 1.1% CER improvement observed due to the synergy when both components are combined highlights their complementary roles: the BLOCKPROCESSOR enables richer feature extraction, while the Modified Text Tokenizer ensures compact and accurate Arabic text representation. This shows the importance of aligning task-specific components to the target language, as their interaction can yield significant synergistic effects beyond individual contributions.

**Synthetic Stage-1 Fine-Tuning (E).** Removing the synthetic Stage-1 fine-tuning resulted in a substantial increase in CER by 4.2%. This demonstrates the effectiveness of the Stage-1 fine-tuning step that allows the model to better address the three inherent challenges of Arabic.

**Pretrained Weights (F).** When the training of HATFORMER was started from randomly initialized weights, the model's performance plummeted to a CER of 86.0%. Despite the major differences between English and Arabic scripts, leveraging TrOCR's synthetic pretraining checkpoint for English OCR led to better results.

**Arabic Specific Postprocessing Normalization.** We leveraged Arabic domain knowledge to group our model errors into normalization categories: replace without context, replace with context, and remove diacritics. The replace without context category normalizes characters to a single form that is phonetically similar and generally does not change the meaning of the word. The replace with context category is where more aggressive normalization is applied. Characters that are similar but can change the word's meaning are converted to a single form. Remove diacritics is relevant to applications such as historical informational archival and search, where normalizing certain characters into a single form is acceptable. Diacritics, in some cases, can be sparsely used and be removed in an Arabic OCR system. Table 4 shows that the model performance in terms of CER improves by 1.9% points or an additional ∼0.6% per post-processing for each category.

## 5.6 FACTOR/SENSITIVITY STUDY

We analyze the impact of various parameters on model performance, with additional experiments in Appendix C.

**Block Processor Methods.** Several studies have explored dynamic aspect ratio image-processing approaches in vision-language models (Bavishi et al., 2023; Fadeeva et al., 2024; Dehghani et al., 2024). We compared our proposed BLOCKPROCESSOR with two notable methods:

Table 5: Block Processor Comparison

| Processor | CER (%) ↓ |
| --- | --- |
| Lee et al. (2023) | 20.3 |
| Li et al. (2023) | 11.4 |
| BLOCKPROCESSOR | **8.6** |

TrOCR (Li et al., 2023), which employs the standard ViT processing approach by resizing input images to 384-by-384 pixels, and Pix2Struct (Lee et al., 2023), which scales input images while preserving the aspect ratio to extract the maximum number of patches within a given sequence length. Table 5 shows that our BLOCKPROCESSOR achieves the best CER of 8.6% on the Muharaf dataset. As discussed in Sections 4.1 and 5.5, TrOCR's processor suffers from information loss due to its inefficient representation of resized images. While Pix2Struct addresses this by preserving the aspect ratio, it introduces variability in the semantic meaning of patches, even when using absolute 2-dimensional positional embeddings. A patch may correspond to a character fragment in shorter images, while a patch might represent an entire character in longer images. This inconsistency in patch representation can negatively impact the model's ability to interpret and process the input.

## 6 CONCLUSION AND LIMITATIONS

In this paper, we have presented HATFORMER, a dedicated Arabic handwritten text recognition system harnessing the transformer's attention mechanism to address the unique challenges of the Arabic language. Our system integrates training methods with image and text processing techniques designed for Arabic HTR. Experiments show that HATFORMER outperforms baseline methods across multiple real-world datasets, highlighting the effectiveness of our approach.

HATFORMER demonstrates significant progress in historical Arabic handwritten text recognition but also has some limitations. As a text line recognition model, its performance relies on the quality of line segmentations during real-world inference. Additionally, HATFORMER struggles with line images exhibiting extreme slants without angle normalization, which can impact recognition accuracy. The computational demands of the training process, particularly with the overtraining strategy, pose challenges for institutions with extremely limited resources. Addressing this, future work could explore parameter-efficient fine-tuning, such as low-rank adaptation (LoRA) (Hu et al., 2022) to enhance accessibility. These limitations point to key areas for improvement, including preprocessing enhancements and optimization of training methods, to increase robustness and applicability across diverse contexts.

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

# A   ATTENTION MAPS

We analyze the effectiveness of using self-attention for Arabic HTR, including visualizing the self-attention maps of our vision transformer encoder. We also visualize the cross-attention maps corresponding to a selected ground truth token.

Our visualization scheme for the vision transformer involves selecting a patch of interest in the image and then visualizing how it attends to other patches. In the heatmaps, a brighter color (yellow) indicates that the selected patch pays more attention to this patch. We accumulate the self-attention heatmaps from the previous layers (taking into account residual connections) in the next layer to get a more holistic view of the attention flow. In the following text, we discuss insights from the attention maps and how our transformer model deals with the intricacies of Arabic handwriting and addresses key Arabic script challenges that we outlined in Section 3.

**Cursive.** Figure 3 demonstrates that when a patch containing a cursive line is selected, the attention map first highlights the relevant strokes of that character. This indicates that the network learns to distinguish individual characters and strokes before applying broader, global attention, effectively connecting relevant patches. The cross-attention map further shows that the model accurately identifies character boundaries. The model successfully segments the entire word in the image for tokens with complex cursive dependencies.

**Context-Sensitive.** A character in Arabic can take multiple forms depending on its position within a word and the surrounding characters. The cross-attention maps in Figure 3 demonstrate that even when words are split into multiple tokens, the model can accurately differentiate between word pieces and segment each part. These maps reveal that the model effectively learns the complex morphological rules of Arabic script and can distinguish between different positional forms of the same character.

**Diacritics.** Figure 3 demonstrates that the diacritic patch can attend to the character patches it is associated with. Importantly, both self-attention and cross-attention maps indicate that diacritic marks are not treated as noise but carry a strong signal. The self-attention maps reveal that the model

can associate the relevant character corresponding with the diacritic. The cross-attention maps show the model correctly identifying the position of diacritics within the corresponding word. These maps highlight the network's ability to incorporate these small marks into the final token predictions rather than ignoring them.

**Attention Maps Cutoff.** To further evaluate the self-attention mechanism, our Arabic expert coauthor analyzed the progression of attention associations across layers. Specifically, our expert identified layers where the attention between a patch and other patches becomes excessively broad relative to the associated word, potentially diluting the model's focus on relevant features. These cutoff points are marked with red lines in the visualizations. This analysis provides valuable insights into how effectively the model maintains meaningful associations and highlights potential areas for improvement, particularly in leveraging Arabic-specific linguistic and structural knowledge.

## B DATASETS

### B.1 MUHARAF

The Muharaf dataset (Saeed et al., 2024) is a public collection of historical handwritten Arabic manuscripts spanning from the early 19th century to the early 21st century. The dataset encompasses diverse document types, including personal letters, poems, dialogues, legal records, correspondences, and church documents. It consists of over 36,000 text line images, exhibiting significant variability in quality. These range from clear handwriting on clean white paper to highly degraded illegible text on creased pages with ink bleed-through. Fluent Arabic speakers scanned and transcribed the historical documents, ensuring high-quality annotations. This makes the Muharaf dataset a valuable resource for advancing research in historical handwriting recognition in Arabic.

### B.2 KHATT

The KHATT dataset (Mahmoud et al., 2012) is a standard benchmark for Arabic HTR tasks. It is a public collection of modern Arabic handwriting samples comprising over 6,600 segmented line images. All images feature black text written on clean white backgrounds, ensuring consistent visual quality. The dataset was created under controlled conditions, where 1,000 participants transcribed 2,000 unique texts provided to them.

### B.3 MADCAT

The MADCAT dataset (Lee et al., 2012; 2013a;b) is a proprietary dataset created by the Linguistic Data Consortium (LDC) to support the DARPA MADCAT Program. It comprises 740,000 modern handwritten Arabic line images created under controlled conditions with standardized writing speed, methodology, tool, and paper-type specifications. The text content was sourced from various digital mediums, including weblogs, newswires, and newsgroups. Each image features black text on a clean white background, ensuring high visual consistency. Due to its large size, the MADCAT dataset is a valuable resource for advancing Arabic HTR research.

### B.4 SYNTHETIC

For our Stage 1 training dataset, we generated 1,000,065 synthetic images of Arabic text lines. To create these, we randomly sampled between 1 and 20 words inclusive from an Arabic corpus comprising 8.2 million words, constructed by combining the datasets from Abbas & Smaili (2005); Abbas et al. (2011); Saad & Alijla (2017). The selected words were rendered using one of 54 Arabic fonts and placed on a background randomly selected from a set of 130 paper background textures. We source the Arabic fonts from freely available online websites. The 130 paper backgrounds are created from the Muharaf dataset by copying parts of the background image or created by using

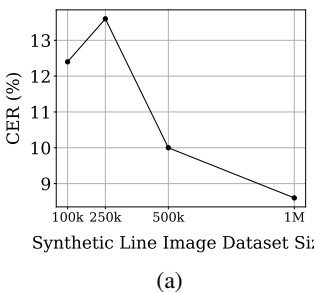 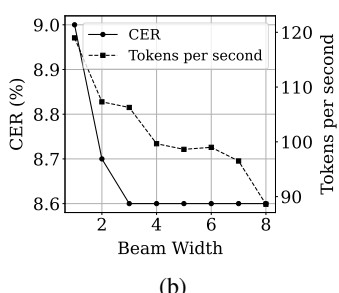 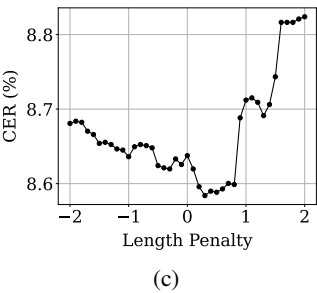

|     |     |     |
| --- | --- | --- |
| (a) | (b) | (c) |

Figure 4: (a) The impact of synthetic Stage-1 fine-tuning size on final HTR performance. A larger synthetic Stage-1 fine-tuning dataset allows for better generalization in terms of CER. (b) The CER and latency effect of inference beam size of our model on Muharaf. Using a larger beam size leads to a more accurate model but reduced speed. A beam width of three demonstrates a good trade-off between accuracy and computational speed. (c) The impact of inference length penalty of our model on Muharaf. A length penalty of 0.2 to 0.8 is preferred to achieve the best CER.

online paper texture images. Additionally, we applied one of eight image augmentation techniques: width distortion, height distortion, barrel distortion, left arc, right arc, left rotation, right rotation, or no distortion. We will release the realistic Arabic synthetic dataset and code to generate the images.

## C  ADDITIONAL FACTOR/SENSITIVITY STUDY

We analyze the impact of various parameters on model performance with further discussion and comparison of image processors in Section 5.6. [1]

**Stage-1 Synthetic Dataset Size.** Figure 4(a) shows the impact of the synthetic Stage-1 training dataset size on the final performance of HATFORMER on the Muharaf dataset. As the size of the synthetic dataset increases, the CER decreases, demonstrating improved generalization. Specifically, datasets of 500k images and 1M images yield the best performance, with the CER dropping below 10%. This trend suggests that a larger synthetic Stage-1 training dataset enhances the model's ability to effectively handle the inherent challenges of Arabic, ultimately leading to better CER performance in downstream HTR tasks.

**Consecutive Whitespaces.** The reported evaluation results throughout this paper were derived by removing consecutive whitespaces at test time. This is in line with the default CER score implementation in the HuggingFace evaluate library (v0.4.2). We observed that performing this normalization during the training stage instead of inference time leads to an additional 0.2% CER improvement.

**Inference Beam Width.** Figure 4(b) shows the effect of beam width on CER and generation speed. The CER improves until the beam width is three and stabilizes beyond this point. Hence, we used a beam width of three in our reported results. The inference speed was measured in tokens per second over the Muharaf test set (total time / total tokens) on a single A10 (24GB) GPU with a batch size of 1. From the inference speeds we can see that our model can be used in a low-resource environment.

**Inference Length Penalty.** The length penalty parameter in beam search biases the generated output sequence length, where negative values encourage shorter sequences and positive values encourage longer ones. In Figure 4(c), we empirically show that on the Muharaf dataset, our model performs optimally using a length penalty between 0.2 and 0.8.

---

[1]One notable parameter that was infeasible to study was decreasing the ViT patch size due to training computational complexity. Reducing patch size in ViT's results in a longer sequence and the attention mechanism requires quadratic cost $O(n^2)$ with respect to sequence length $n$.

