# OpenReview forum: "HATFormer: Historic Handwritten Arabic Text Recognition with Transformers"
_ICLR.cc/2025/Conference — Submitted to ICLR 2025_

### Official Review · Reviewer_aDtf · 2024-10-28

**Soundness:** 2
**Presentation:** 3
**Contribution:** 2
**Rating:** 5
**Confidence:** 4

**Summary:**

This paper describes a model for handwritten OCR of historic texts in Arabic script. The proposed model uses finetunes the TrOCR approach, first on a large synthetic and then on a small real dataset. To feed the large non-square images to the model, they are written in multiple lines when presented to the vision encoder. The model seems to obtain state-of-the-art results on 2/4 datasets and shows strong performance in cross-dataset evaluation.

**Strengths:**

The paper describes the approach to obtaining state-of-the art performance in a particular domain of OCR on historic datasets in Arabic script. The writing is clear and the approach seems to be easy to reproduce.

**Weaknesses:**

Novelty: The main novelty cited in the paper are the BlockProcessor and Arabic BBPE tokenizer. However, block processing of the images for transformer vision encoder has already been studied in several works (ex. Fuyu-8B in https://www.adept.
ai/blog/fuyu-8b or "Representing Online Handwriting for Recognition in Large Vision-Language Models", Fadeeva et al.) and the BPE tokenizer is a standard solution for VLMs. Furthermore, according to Table 3, the use of BlockProcessor actually decreases the model performance. The paper could be strengthened by offering a comparison to those approaches, or to other approaches targeted at solving a similar problem differently (ex. Pack'n'Patch in https://arxiv.org/abs/2307.06304 or Pix2Struct in https://arxiv.org/abs/2210.03347)

Beyond the aforementioned improvements, the proposed approach performs a (two-stage) fine-tuning of the base model, which does yield good performance, but may not be a significant enough contribution in itself. The paper could be strengthened by releasing the model weights or training code, which would be an additional contribution. Alternatively, perhaps open-sourcing some parts, like the synthetic data generation pipeline or dataset, could also be a good contribution. UPD: authors provided the link to their model, and I am increasing the rating based on that.

**Questions:**

I don't have particular questions to the authors but I believe that one way to strengthen the manuscript could be to open-source the model. UPD: authors provided the link to their model, and I am increasing the rating based on that.

---

> ### Author Response · Authors · 2024-11-24
> **Response to Reviewer aDtf (Part 1)**
>
> >**Q1: Novelty: The main novelty cited in the paper are the BlockProcessor and Arabic BBPE tokenizer. However, block processing of the images for transformer vision encoder has already been studied in several works and the BPE tokenizer is a standard solution for VLMs.**
>
> We appreciate the reviewer’s insightful comments regarding existing block processing methods.  While we acknowledge that block processing for transformer vision encoders and BPE tokenization are established techniques, our contributions lie in adapting these methods specifically for Arabic handwriting recognition.
>
> **[Edit as of 11/25 12:10 AM EST]**
>
> We thank the reviewer for their patience and thoughtful feedback. To address the concerns raised, we conducted a comparative experiment between our BlockProcessor and the Pix2Struct processor (Lee et al., 2023), as it aligns closely with our encoder-decoder architecture and has publicly available code for implementation. We have updated Table 5 and analysis in Section 5.6 to include our comparison results with Pix2Struct. This additional analysis demonstrates the strengths of our approach in handling Arabic handwritten text. We decided not to include a comparison with Fuyu-8B's image processing method because it is based on a decoder-only architecture, which fundamentally differs from our encoder-decoder approach. Regarding Fadeeva et al. (2024), while their multi-line rendering method shares some similarities with our work, they focus on modern online handwriting, outside the scope of our focus on historical Arabic handwritten text. Additionally, Pack'n'Patch (Dehghani et al., 2024) lacks publicly available code, which limits our ability to perform a fair and reproducible comparison.
>
> The updated table and description under Section 5.6 on page 10 are as follows:
> | Processor | CER |
> |---|---|
> Lee et al. (2023) | 20.3
> Li et al. (2023) | 11.4
> BlockProcessor | **8.6**
>
> Block Processor Methods. Several studies have explored dynamic aspect ratio image-processing approaches in vision-language models (Bavishi et al., 2023; Fadeeva et al., 2024; Dehghani et al., 2024). We compared our proposed BlockProcessor with two notable methods: TrOCR (Li et al., 2023), which employs the standard ViT processing approach by resizing input images to 384-by-384 pixels, and Pix2Struct (Lee et al., 2023), which scales input images while preserving the aspect ratio to extract the maximum number of patches within a given sequence length. Table 5 shows that our BlockProcessor achieves the best CER of 8.6% on the Muharaf dataset. As discussed in Section 4.2 and Section 5.5, TrOCR's processor suffers from information loss due to its inefficient representation of resized images. While Pix2Struct addresses this by preserving the aspect ratio, it introduces variability in the semantic meaning of patches, even when using absolute 2-dimensional positional embeddings. A patch may correspond to a character fragment in shorter images, while a patch might represent an entire character in longer images. This inconsistency in patch representation can negatively impact the model's ability to interpret and process the input.

---

> ### Author Response · Authors · 2024-11-25
> **Response to Reviewer aDtf (Part 2)**
>
> **[Edit as of 11/25 12:10 AM EST]: Moved Question 2 and 3 down for clarity**
>
> >**Q2: Furthermore, according to Table 3, the use of BlockProcessor actually decreases the model performance.**
>
> We thank the reviewer for their feedback. However, we would like to clarify that using our proposed BlockProcessor has consistently improved our model performance in all our experiments. Row (C1), formerly (C), showed the performance of the modified BBPE tokenizer without our proposed BlockProcessor. From row (C1), formerly (C), to row (D) shows the performance change when replacing the modified BBPE tokenizer with TrOCR’s tokenizer. We additionally identified a bug, which has now been fixed. The corrected results are provided in Table 3, and we have expanded the discussion on page 9 to provide further clarity. Specifically, the 21.3% CER reported in row C of the original manuscript was incorrect, and it has now been updated to 11.4% CER in row C1.
>
> In response to the reviewer’s feedback, we conducted two ablation studies to validate our results. Row C1 presents a repeated experiment from the original manuscript (i.e., replacing the BlockProcessor with the TrOCR Processor), while row C2, which is newly added, shows the effects of replacing the custom Arabic BBPE tokenizer with the TrOCR tokenizer.
>
> The updated table and description are as follows:
> | |Model|CER|
> |---|---|---|
> (A) | Proposed Model | 8.6
> (B) | (A) - Overtraining | 9.9
> **(C1)** | **(B) - BlockProcessor + TrOCR Processor** | **11.4**
> **(C2)** | **(B) - Modified Text Tokenizer + TrOCR Tokenizer** | **10.0**
> (D) | (B) - (C1) - (C2) | 10.4
> (E) | (D) - Synthetic Stage-1 fine-tuning | 14.6
> (F) | (E) - Pretrained Weights | 86.0
>
> “(C1) & (C2) & (D) BlockProcessor and Modified Text Tokenizer. When the BlockProcessor and Arabic BBPE were added together, this led to a 0.5% CER improvement. This supports our ideas in Sections 4.1 and 4.2 and shows the effectiveness of our approach. Replacing TrOCR’s image processor with our BlockProcessor led to a 0.4% CER improvement, whereas replacing the modified text tokenizer with TrOCR’s tokenizer led to a -1.0% performance change. This indicates that the BlockProcessor enhances image feature extraction. However, the modified text tokenizer struggles when paired with TrOCR’s processor due to the naive resizing, which discards essential features needed for predicting compact Arabic token representations and achieving accurate HTR, as discussed in Section 4.1. The 1.1% CER improvement observed due to the synergy when both components are combined highlights their complementary roles: the BlockProcessor enables richer feature extraction, while the Modified Text Tokenizer ensures compact and accurate Arabic text representation. This shows the importance of aligning task-specific components to the target language, as their interaction can yield significant synergistic effects beyond individual contributions."
>
> >**Q3: The paper could be strengthened by releasing the model weights or training code, which would be an additional contribution. Alternatively, perhaps open-sourcing some parts, like the synthetic data generation pipeline or dataset, could also be a good contribution.**
>
> The script for training and making inferences with HATFormer is already available on the discussion forum, and we are committed to publicly releasing all our code upon publication. As outlined in our introduction, we plan to share our “image preprocessor, tokenizer, model weights, and source code for our HTR system.” Additionally, we will release “our dataset of realistic synthetic Arabic images and its generation source code, a detailed guide to help researchers integrate our system with existing text detection packages for page-level HTR evaluations and practical deployment, as well as provide an OCR Error Diagnostic App.” We believe these resources will significantly benefit both machine learning and historical research. We hope these efforts will enhance the reproducibility and accessibility of our work. We would like to thank the reviewer for highlighting the importance of this contribution.

---

> ### Author Response · Authors · 2024-11-29
> **Gentle Reminder**
>
> Dear Reviewer aDtf,
>
> This is a gentle reminder as the deadline is approaching. We would love to hear from you on our rebuttal to see whether we have resolved your concerns, and please let us know if you have any additional questions.
>
> Best Regards, Authors

---

> > ### Comment · Reviewer_aDtf · 2024-12-02
> >
> > Dear Authors,
> >
> > Thank you for your reply, and especially for providing links to the code available. I update the "contribution" and overall rating by 1.

---

### Official Review · Reviewer_KDgu · 2024-11-01

**Soundness:** 3
**Presentation:** 3
**Contribution:** 3
**Rating:** 6
**Confidence:** 4

**Summary:**

This manuscript presents a historical handwritten Arabic line recognition based on transformers. This manuscript is well written and organised. This approach is evaluated on some datasets and promising results are reported.

However, there are some problems with the presentation and experimental results.

**Strengths:**

This paper improves the results of the current state of the art in this field.

**Weaknesses:**

- This approach focuses on the extracted line to recognise the documents and this should appear in the title. It is part of a document analysis system and this may lead to errors in the related subsystems.

- It is better to explain the records in detail in the appendix. The reader does not get an idea of which datasets are historical and how the datasets differ from other document analysis datasets. Also, we do not have detailed information about the generated dataset, which should be explained in detail.

- I can see that the authors provide an analysis in the appendix to show the three intrinsic challenges. But that is not enough. They should provide a comprehensive analysis to show how effective the method is in overcoming the challenges.

- I can not see any analysis that shows the limitations of the approach. The flaws of the approach are not shown in the experimintal results.

**Questions:**

My comments are explained in the weaknesses section.

---

> ### Author Response · Authors · 2024-11-24
> **Response to Reviewer KDgu**
>
> >**Q1: This approach focuses on the extracted line to recognise the documents and this should appear in the title. It is part of a document analysis system and this may lead to errors in the related subsystems.**
>
> We appreciate the reviewer’s feedback. Based on the suggestion, we have updated the title to reflect the focus of our approach more accurately. The revised title is:
>
> “HATFormer: Historic Handwritten Arabic Text Line Recognition with Transformers”
>
> We also incorporated this feedback into our limitations paragraph under Section 6 on page 10. Specifically, we add this sentence:
>
> “As a text line recognition model, its performance relies on the quality of line segmentations during real-world inference.”
>
> >**Q2: It is better to explain the records in detail in the appendix. The reader does not get an idea of which datasets are historical and how the datasets differ from other document analysis datasets. Also, we do not have detailed information about the generated dataset, which should be explained in detail.**
>
> We thank the reviewer for highlighting the need for more detailed information on the datasets used. In the revised version, we have added a more comprehensive (¾ of a page) description of the datasets. We refer the reviewer to Appendix B on Page 15 for the updated text due to the large amount of content.
>
> >**Q3: I can see that the authors provide an analysis in the appendix to show the three intrinsic challenges. But that is not enough. They should provide a comprehensive analysis to show how effective the method is in overcoming the challenges.**
>
> We appreciate the reviewer’s suggestion for a more in-depth analysis. In response, we have improved upon Figure 3 and expanded the analysis. We have added the following lines:
> “Red lines indicate the layer cutoff where the attention association becomes too broad, as identified by our Arabic expert.”
> “Tokens are annotated based on their type: red underlines indicate diacritic tokens, green underlines denote subword tokens, and all other tokens correspond to full words, as identified by our Arabic language expert.”
> We have also added a subsection explaining the attention layer cutoffs to Appendix A on line 755:
> “To further evaluate the self-attention mechanism, our Arabic expert coauthor analyzed the progression of attention associations across layers. Specifically, our expert identified layers where the attention between a patch and other patches becomes excessively broad relative to the associated word, potentially diluting the model's focus on relevant features. These cutoff points are marked with red lines in the visualizations. This analysis provides valuable insights into how effectively the model maintains meaningful associations and highlights potential areas for improvement, particularly in leveraging Arabic-specific linguistic and structural knowledge.”
>
> >**Q4: I can not see any analysis that shows the limitations of the approach. The flaws of the approach are not shown in the experimintal results.**
>
> We appreciate the reviewer’s valuable feedback. In response, we have included a dedicated paragraph discussing the limitations of our approach. The new paragraph  under Section 6 on page 10 reads as follows:
>
> “HATFormer demonstrates significant progress in historical Arabic handwritten text recognition but also has some limitations. As a text line recognition model, its performance relies on the quality of line segmentations during real-world inference. Additionally, HATFormer struggles with line images exhibiting extreme slants without angle normalization, which can impact recognition accuracy. The computational demands of the training process, particularly with the overtraining strategy, pose challenges for institutions with extremely limited resources. The computational demands of the training process, particularly with the overtraining strategy, pose challenges for institutions with extremely limited resources. Addressing this, future work could explore parameter-efficient fine-tuning, such as low-rank adaptation (LoRA) (Hu et al., 2022) to enhance accessibility. These limitations point to key areas for improvement, including preprocessing enhancements and optimization of training methods, to increase robustness and applicability across diverse contexts.”

---

> > ### Comment · Reviewer_KDgu · 2024-11-26
> >
> > Dear Authors: Thank you for your responses. I have just updated my rating for the soundness, presentation and contribution items. I am keeping my general rating the same for the round.

---

### Official Review · Reviewer_8c8C · 2024-11-01

**Soundness:** 3
**Presentation:** 3
**Contribution:** 2
**Rating:** 5
**Confidence:** 2

**Summary:**

This paper presents a transformer based method, HATFormer, to analyze historic handwritten Arabic. The authors proposed the BlockProcessor that effectively allows the vision transformer to learn from Muharaf Arabic dataset. The method beats existing CRNN methods in the Muharaf dataset.

**Strengths:**

* The authors described the BlockProcessor which helps preprocess the handwritten texts images for the ViT, substantially improves the performance. Although the authors applied this processor to Arabic, it could potentially help with other scripts
* The proposed model had significant improvement in the Muharaf dataset compared to existing methods (CER=17.6 -> 8.6)
* The model trained with the Muharaf dataset could also be generalised to other datasets

**Weaknesses:**

* The authors emphasized the importance of pre-training with synthetic data however details regarding of this pre-training procedures was very limited.
* The authors mentioned the model was optimised for the Muharaf dataset, but, only provided one existing method for comparisons using this dataset

**Questions:**

* When you retrained the Saeed 2024 model for evaluation, did you also perform pretraining with synthetic printed data?

---

> ### Author Response · Authors · 2024-11-24
> **Response to Reviewer 8c8C**
>
> >**Q1: The authors emphasized the importance of pre-training with synthetic data however details regarding of this pre-training procedures was very limited.**
>
> We thank the reviewer for raising this important point. We would like to clarify that the details of the synthetic pretraining procedure are discussed in Section 5.2 on lines 346 and 349. We provide details on the synthetic pre-training procedure, including the “90–9–1” data split and “traditional validation loss as the early stopping criterion, with a maximum of 5 epochs” for Stage 1 training. Additionally, we describe the synthetic dataset in Section 5.1 and show a diagram of this pipeline in Figure 2(g) on page 5. Specifically, the dataset consists of “1,000,065 synthetic images of Arabic text lines”, with words sampled from a corpus and “paired with one of 54 Arabic fonts.” These words were placed on “backgrounds chosen from 130 paper images, and we applied one of eight image augmentation techniques to generate the synthetic line images.”
>
> We also add the following to Appendix B.4 for more details on the synthetic dataset:
>
> “For our Stage 1 training dataset, we generated 1,000,065 synthetic images of Arabic text lines. To create these, we randomly sampled between 1 and 20 words inclusive from an Arabic corpus comprising 8.2 million words, constructed by combining the datasets from Abbas & Smaili (2005); Abbas et al. (2011); Saad & Alijla (2017). The selected words were rendered using one of 54 Arabic fonts and placed on a background randomly selected from a set of 130 paper background textures. We source the Arabic fonts from freely available online websites. The 130 paper backgrounds are created from the Muharaf dataset by copying parts of the background image or created by using online paper texture images. Additionally, we applied one of eight image augmentation techniques: width distortion, height distortion, barrel distortion, left arc, right arc, left rotation, right rotation, or no distortion. We will release the realistic Arabic synthetic dataset and code to generate the images.”
>
> >**Q2: The authors mentioned the model was optimised for the Muharaf dataset, but, only provided one existing method for comparisons using this dataset**
>
> The Muharaf dataset is relatively new, and currently, only one existing method in the literature has been evaluated on it, specifically, the approach proposed by Saeed et al. (2024). We clarified this on line 364 to avoid any confusion regarding the availability of additional comparative methods for the Muharaf dataset and have provided the revised paragraph below:
>
> “Table 1 reports the CER for HATFormer and several existing baselines across the three datasets. An important note is that the only existing baseline for the Muharaf dataset is Saeed et al. (2024). Since the source code for many existing Arabic HTR baseline models is not publicly available, except Saeed et al. (2024), we compared our results to the reported numbers obtained from their papers.”
>
> >**Q3: When you retrained the Saeed 2024 model for evaluation, did you also perform pretraining with synthetic printed data?**
>
> Yes, we did perform pretraining with synthetic printed data when retraining the model from Saeed et al. (2024) for evaluation. We clarify this by editing the following sentence on line 368:
>
> “For Saeed et al. (2024), we retrained their model on each dataset and with stage-1 synthetic training for a fair comparison.”

---

> ### Author Response · Authors · 2024-11-29
> **Gentle Reminder**
>
> Dear Reviewer 8c8C,
>
> This is a gentle reminder as the deadline is approaching. We would love to hear from you on our rebuttal to see whether we have resolved your concerns, and please let us know if you have any additional questions.
>
> Best Regards, Authors

---

> ### Author Response · Authors · 2024-12-03
> **Gentle Reminder**
>
> Dear Reviewer 8c8C,
>
> We kindly request your feedback to see if our response and revisions effectively resolve your concerns. We hope that you find our responses convincing. Please let us know if you have any additional questions.
>
> Best Regards, Authors

---

### Official Review · Reviewer_EpDq · 2024-11-02

**Soundness:** 2
**Presentation:** 2
**Contribution:** 2
**Rating:** 6
**Confidence:** 5

**Summary:**

This paper presents an improvement on the transformer-based encoder-decoder architecture of the TrOCR OCR model. These improvements are designed to address the challenges of Arabic handwritten text recognition, particularly for historical texts. The architecture includes a custom image processor for ViT information pre-processing, a specialized text tokenizer for Arabic text representation, and a training pipeline optimised for limited historical Arabic data. The system is compared with several other HTR systems based on 3 public databases and generally outperforms them.

**Strengths:**

- Original use of line warping: The paper presents an innovative approach using line warping techniques to adapt elongated rectangular text lines into a square format suitable for Vision Transformers (ViTs). This adaptation is a solution to the challenges posed by the non-uniformity and the specific shape of handwritten text.

- Extensive pre-training on Arabic data: HATFORMER benefits from extensive pre-training on a synthetic Arabic dataset, which is critical given the limited availability of such data compared to English datasets. Note that using the English pre-training is still beneficial as showed in Table3

- Performance: The model outperforms a state-of-the-art model on one dataset, and is on par on 2 other datasets.
- Open-source : the code and dataset will be released

**Weaknesses:**

Ablation study results: The ablation study presents some unusual results that need further investigation. The finding that the removal of block processing leads to a significant performance degradation suggests that the novel image packing technique is beneficial. However, the fact that removing the custom byte pair encoding (BBPE) improves the results - almost to the level of the original model - raises questions. This behaviour is counterintuitive and should be investigated further. The authors should conduct additional experiments to clarify these results and rule out the possibility of bugs or unintended interactions within the model.

Generalisability: Although the aim is to develop a generic HTR model, the system has not been trained on all available datasets to assess its generalisation ability. Testing this generic model on different datasets would provide valuable insights into its robustness and adaptability to different handwriting styles and historical contexts.

**Questions:**

L56 : "First, Arabic is written in cursive, making characters visually harder to distinguish"  English is also cursive
L68 : The number of handwritten pages in the RIMES data base is not 12500, because only a fraction of the database is fully handwritten
The OpenHART database is not mentioned in this section but it is composed of more than 40 000 annotated pages of handwritten Arabic https://catalog.ldc.upenn.edu/LDC2012T15. However, it is used in the experiments.


Related work section

References to full page recognition (for example DAN Coquenet, D., Chatelain, C., Paquet, T.: DAN: a segmentation-free document attention network for handwritten document recognition. IEEE Trans. Pattern Anal. Mach. Intell. 45(7), 8227–8243 (2023). https://doi.org/10.1109/TPAMI.2023.3235826)  are missing


L200 : "the proposed BLOCKPROCESSOR works by first horizontally flipping a text-line image"  : since the usage of the pre-trained English weights is capital, we can see that changing the order of the patches in the Vit would be doomed to failure. However, the approach of returning the image of the line is not very satisfactory because, on the one hand, it does not adapt the model to the specific characteristics of Arabic, even though the authors' objectives are to take these specific characteristics into account, and on the other hand, it poses serious problems in the event of mixing text in Arabic and Latin alphabets.

L201 : "finally warping it to fill in the ViT’s 384×384-pixel image container from left to right and top to bottom" : this means that the system could decode paragraphs

---

> ### Author Response · Authors · 2024-11-24
> **Response to Reviewer EpDq (Part 1)**
>
> >**Q1: The ablation study presents some unusual results that need further investigation. The finding that the removal of block processing leads to a significant performance degradation suggests that the novel image packing technique is beneficial. However, the fact that removing the custom byte pair encoding (BBPE) improves the results - almost to the level of the original model - raises questions. This behaviour is counterintuitive and should be investigated further. The authors should conduct additional experiments to clarify these results and rule out the possibility of bugs or unintended interactions within the model.**
>
> We thank the reviewer for pointing out the unusual findings in our ablation study. Upon investigation, we identified a bug, which has now been fixed. The corrected results are provided in Table 3, and we have expanded the discussion on page 9 to provide further clarity. Specifically, the 21.3% CER reported in row C of the original manuscript was incorrect, and it has now been updated to 11.4% CER in row C1.
>
> In response to the reviewer’s suggestions, we conducted two ablation studies to validate our results. Row C1 presents a repeated experiment from the original manuscript (i.e., replacing the BlockProcessor with the TrOCR Processor), while row C2, which is newly added, shows the effects of replacing the custom Arabic BBPE tokenizer with the TrOCR tokenizer.
>
> The updated table and description are as follows:
> | |Model|CER|
> |---|---|---|
> (A) | Proposed Model | 8.6
> (B) | (A) - Overtraining | 9.9
> **(C1)** | **(B) - BlockProcessor + TrOCR Processor** | **11.4**
> **(C2)** | **(B) - Modified Text Tokenizer + TrOCR Tokenizer** | **10.0**
> (D) | (B) - (C1) - (C2) | 10.4
> (E) | (D) - Synthetic Stage-1 fine-tuning | 14.6
> (F) | (E) - Pretrained Weights | 86.0
>
> “(C1) & (C2) & (D) BlockProcessor and Modified Text Tokenizer. When the BlockProcessor and Arabic BBPE were added together, this led to a 0.5% CER improvement. This supports our ideas in Sections 4.1 and 4.2 and shows the effectiveness of our approach. Replacing TrOCR’s image processor with our BlockProcessor led to a 0.4% CER improvement, whereas replacing the modified text tokenizer with TrOCR’s tokenizer led to a -1.0% performance change. This indicates that the BlockProcessor enhances image feature extraction. However, the modified text tokenizer struggles when paired with TrOCR’s processor due to the naive resizing, which discards essential features needed for predicting compact Arabic token representations and achieving accurate HTR, as discussed in Section 4.1. The 1.1% CER improvement observed due to the synergy when both components are combined highlights their complementary roles: the BlockProcessor enables richer feature extraction, while the Modified Text Tokenizer ensures compact and accurate Arabic text representation. This shows the importance of aligning task-specific components to the target language, as their interaction can yield significant synergistic effects beyond individual contributions.”
>
> >**Q2:  Generalisability: Although the aim is to develop a generic HTR model, the system has not been trained on all available datasets to assess its generalisation ability. Testing this generic model on different datasets would provide valuable insights into its robustness and adaptability to different handwriting styles and historical contexts.**
>
> In response to the reviewer’s suggestion, we have conducted further experiments combining the Muharaf, KHATT, and MADCAT datasets. We have added the results to Table 1 under Section 5. This additional analysis further supports the robustness of our approach.
>
> |Dataset|Model|Architecture|CER
> |---|---|---|---|
> |...|...|...|...|
> |**Combined**|**Proposed Model**|**Transformer**|**15.3**|
>
> Below is the additional explanation of our results added to line 447:
>
> “We combined the three handwritten datasets into a single large dataset to evaluate the model's performance across diverse handwriting styles. Using this combined dataset, HATFormer achieved a CER of 15.3%. While this result is slightly worse higher than the individual dataset CERs, it reflects the challenge of adapting to significant image content and style variability across the Muharaf, KHATT, and MADCAT datasets, indicating that HATFormer can still extract meaningful shared features even with the increased difficulty of combining datasets.”

---

> > ### Comment · Reviewer_EpDq · 2024-11-30
> > **Evaluation of the combined model**
> >
> > thank you for adding the evaluation of the combined model. Could you also add its result on each database on Table 2 ?
> >
> > I am happy to see that the discussion has permitted to find a bug in your code.
> >
> > I raise my evaluation.

---

> ### Author Response · Authors · 2024-11-24
> **Response to Reviewer EpDq (Part 2)**
>
> >**Q3: L56 : "First, Arabic is written in cursive, making characters visually harder to distinguish" English is also cursive**
>
> We appreciate the reviewer’s observation. To clarify, our statement highlights that Arabic script is inherently cursive in its printed and handwritten forms, where the majority of characters connect seamlessly within words. In contrast, English is typically written in a non-cursive style in its printed form, and cursive handwriting is less common or optional. This inherent cursiveness in Arabic contributes to challenges in distinguishing characters, as their shapes can vary significantly depending on their position within a word (initial, medial, final, or isolated). We have revised the wording in the manuscript to reflect this distinction more accurately. The revised line on L56 now reads:
>
> “First, Arabic is required to be written in cursive, making characters visually harder to distinguish.”
>
> >**Q4: L68 : The number of handwritten pages in the RIMES data base is not 12500, because only a fraction of the database is fully handwritten The OpenHART database is not mentioned in this section but it is composed of more than 40 000 annotated pages of handwritten Arabic https://catalog.ldc.upenn.edu/LDC2012T15. However, it is used in the experiments.**
>
> We thank the reviewer for pointing out the inaccuracies regarding the RIMES database and for bringing attention to the MADCAT dataset. We have revised the paragraph to correct these details and incorporate the information about MADCAT. The updated text on line 68 now includes “RIMES (Grosicki et al., 2024), which comprises a mix of handwritten and printed text across approximately 12,500 pages” and mentions the “MADCAT (Lee et al., 2012; 2013a;b) dataset, which contains over 40,000 pages” although it is “not focused on historical writing.”
>
> >**Q5: References to full page recognition (for example DAN Coquenet, D., Chatelain, C., Paquet, T.: DAN: a segmentation-free document attention network for handwritten document recognition. IEEE Trans. Pattern Anal. Mach. Intell. 45(7), 8227–8243 (2023). https://doi.org/10.1109/TPAMI.2023.3235826) are missing**
>
> We appreciate the reviewer highlighting the omission of references to full-page recognition approaches. We have updated the manuscript to include this information. Specifically, we have added the following text to Line 107:
>
> “Newer approaches (Michael et al., 2019; Wang et al., 2020) attempt to incorporate the attention mechanism  (Bahdanau et al., 2015) into the HTR pipeline. Coquenet et al. (2023) use the attention mechanism to perform full-page HTR, bypassing the need for line-level segmentation.”

---

> ### Author Response · Authors · 2024-11-24
> **Response to Reviewer EpDq (Part 3)**
>
> >**Q6: L200 : "the proposed BlockProcessor works by first horizontally flipping a text-line image" : since the usage of the pretrained English weights is capital, we can see that changing the order of the patches in the Vit would be doomed to failure. However, the approach of returning the image of the line is not very satisfactory because, on the one hand, it does not adapt the model to the specific characteristics of Arabic, even though the authors' objectives are to take these specific characteristics into account, and on the other hand, it poses serious problems in the event of mixing text in Arabic and Latin alphabets**
>
> We thank the reviewer for raising these important concerns regarding the BlockProcessor and its adaptation to Arabic script and mixed-language text. Below, we address each point in detail:
>
> **Point 1: “since the usage of the pretrained English weights is capital, we can see that changing the order of the patches in the Vit would be doomed to failure”**
>
> Flipping the line image ensures the scanning order of the ViT follows the same direction as when it was pretrained on English (left to right). This way, smaller index patches still correspond to the first predicted character, and the larger index patches correspond to the last predicted character. Importantly, the key information gained from pretrained English weights is still transferable to Arabic. Specifically, in both languages, the text is written in the middle of the line, and there is a contrast between the handwritten foreground and background of the paper.
>
> **Point 2: “the approach of returning the image of the line is not very satisfactory because, on the one hand, it does not adapt the model to the specific characteristics of Arabic, even though the authors' objectives are to take these specific characteristics into account”**
>
> Our technique of wrapping the line images is meant as a general technique to preserve the image's aspect ratio. In our application to Arabic HTR, it is imperative that we avoid horizontal information loss, such as the clarity of vertical strokes, as described in section 4.1 and Figure 2. Since a characteristic of Arabic is that it is read from right to left, we adapt our model by flipping the line image. Another characteristic of Arabic is that it is a low-resource language with limited training data. The flipping of the line image is beneficial to use to address both these characteristics because we start from pretrained English weights. English is read from left to right, so we also choose to read Arabic from left to right by flipping the line image. As described in Section 4.1, this allows for more efficient transformer training by not putting an additional burden on relearning the position embeddings. The three specific characteristics of the Arabic language we discuss are primarily addressed using the attention mechanism, as we highlight in Appendix A.
>
> **Point 3: “it poses serious problems in the event of mixing text in Arabic and Latin alphabets”**
>
> Our model is intended to be used on Arabic data, so we chose to flip the line image to help with efficient Arabic training, as described in point 2. Flipping an image that contains Latin is reasonable since we have training labels. We still show that our model can significantly outperform Saeed et al. (2024) by 21% in Section 5.3 on the Muharaf (Full) dataset, which has occasional Latin characters.
>
> >**Q7: "finally warping it to fill in the ViT’s 384×384-pixel image container from left to right and top to bottom" : this means that the system could decode paragraphs**
>
> We thank the reviewer for their observation. The system could theoretically decode paragraphs; however, our work focuses on line-level images. Extending the system to paragraphs would require addressing challenges such as greater variability in paragraph lengths compared to individual text lines. We leave the extension of working with paragraphs for future work.

---

> ### Author Response · Authors · 2024-11-29
> **Gentle Reminder**
>
> Dear Reviewer EpDq,
>
> This is a gentle reminder as the deadline is approaching. We would love to hear from you on our rebuttal to see whether we have resolved your concerns, and please let us know if you have any additional questions.
>
> Best Regards, Authors

---

> ### Author Response · Authors · 2024-12-02
> **Response to Reviewer EpDq**
>
> We thank the reviewer for their suggestions. In response, we have conducted additional evaluations of the combined model and have included the results below. Since the PDF editing deadline has passed, we plan to incorporate them into the camera-ready version should the paper be accepted.
>
> |Training Data|Test Data|CER|
> |---|---|---|
> |KHATT| |40.7|
> |MADCAT| |41.4|
> |Muharaf|Muharaf|8.6|
> |Combined| |17.8|
> ||||
> |Muharaf| |27.5|
> |MADCAT| |16.3|
> |KHATT|KHATT|15.4|
> |Combined| |15.0|
> ||||
> |Muharaf| |26.5|
> |KHATT| |18.1|
> |MADCAT|MADCAT|4.2|
> |Combined| |11.5|
>
> Table 6 in the Appendix reports the extra performance gains when the combined dataset is used for training. While training on the combined dataset allows HATFormer to capture shared features and address significant stylistic variability, in two out of the three datasets, the CER is better when trained on a single dataset from the same source compared to the combined dataset. This indicates primarily higher gains from model personalization than from model generalization (through combining multiple training sources) in this specific HTR application scenario. This could be an interesting research question to further explore in theoretical machine learning.

---

### Author Response · Authors · 2024-11-14
**HATFormer Code and Data**

Here is the code and data for HATFormer. It includes the code for the model training and inference, synthetic data generation resources and scripts, and OCR diagnostic tool.

https://doi.org/10.5281/zenodo.13936253

---

### Author Response · Authors · 2024-11-24
**Rebuttal by Authors**

We thank all the reviewers for their valuable feedback and thorough critiques of our manuscript. We have responded point-by-point to each question raised in the reviews. All changes to the revised manuscript are highlighted in blue for easy reference.

---

### Meta-Review · Area_Chair_v4yW · 2024-12-19

**Metareview:**

The paper introduces an approach using line warping techniques to adapt elongated rectangular text lines into a square format suitable for ViTs, addressing challenges posed by the non-uniformity of handwritten text. It benefits from extensive pre-training on a synthetic Arabic dataset, which is crucial given the limited availability of such data. English pre-training also proves beneficial. The model outperforms a state-of-the-art model on one dataset and performs on par with two other datasets. The code and dataset will be made publicly available.

However, the proposed approach involves a two-stage fine-tuning of the base model, which yields good performance but may not be a significant enough contribution on its own. While the authors provide an analysis in the appendix to show three intrinsic challenges, it is not enough. A comprehensive analysis is needed to demonstrate how effective the method is in overcoming these challenges. There is no analysis showing the limitations of the approach. The flaws are not highlighted in the experimental results.

**Additional Comments On Reviewer Discussion:**

Despite the authors addressing the reviewers' questions and highlighting the novelties of the paper, and even though the sub-scores were increased, the average overall rating remains below the acceptance threshold.

---

### Decision · Program_Chairs · 2025-01-22

Reject